# Revised estimates of leprosy disability weights for assessing the global burden of disease: A systematic review and individual patient data meta-analysis

Shri Lak Nanjan Chandran[1], Anuj Tiwari[1], Anselmo Alves Lustosa[2], Betul Demir[3], Bob Bowers[4], Rachel Gimenes Rodrigues Albuquerque[5], Renata Bilion Ruiz Prado[6], Saba Lambert[7], Hiroyuki Watanabe[8], Juanita Haagsma[1], Jan Hendrik Richardus[1]*

1 Department of Public Health, Erasmus MC, University Medical Center Rotterdam, Rotterdam, The Netherlands, 2 Federal University of Piauí, Piauí, Brazil, 3 Department of Dermatology, Firat University Hospital, Elazig, Turkey, 4 The Leprosy Mission International- Bangladesh, Dhaka, Bangladesh, 5 Department of Psychobiology, Universidade Federal de São Paulo, São Paulo, Brazil, 6 Technical Rehabilitation Team, Instituto Lauro de Souza Lima, Bauru, São Paulo, Brazil, 7 Department of Infectious and Tropical Diseases, London School of Hygiene and Tropical Medicine, London, United Kingdom, 8 Niigata College of Nursing, Joetsu, Japan

* j.richardus@erasmusmc.nl

## Abstract

### Background

Leprosy is a chronic bacterial infection caused by *Mycobacterium leprae*, which may lead to physical disability, stigma, and discrimination. The chronicity of the disease and disabilities are the prime contributors to the disease burden of leprosy. The current figures of the disease burden in the 2017 global burden of disease study, however, are considered to be under-estimated. In this study, we aimed to systematically review the literature and perform individual patient data meta-analysis to estimate new disability weights for leprosy, using Health-Related Quality of Life (HRQOL) data.

### Methodology/principal findings

The search strategy included all major databases with no restriction on language, setting, study design, or year of publication. Studies on human populations that have been affected by leprosy and recorded the HRQOL with the Short form tool, were included. A consortium was formed with authors who could share the anonymous individual-level data of their study. Mean disability weight estimates, sorted by the grade of leprosy disability as defined by WHO, were estimated for individual participant data and pooled using multivariate random-effects meta-analysis. Eight out of 14 studies from the review were included in the meta-analysis due to the availability of individual-level data (667 individuals). The overall estimated disability weight for grade 2 disability was 0.26 (95%CI: 0.18–0.34). For grade 1 disability the estimated weight was 0.19 (95%CI: 0.13–0.26) and for grade 0 disability it was 0.13 (95%CI: 0.06–0.19). The revised disability weight for grade 2 leprosy disability is four

**Data Availability Statement:** All relevant data are within the manuscript and its Supporting Information files.

**Funding:** The author(s) received no specific funding for this work.

**Competing interests:** The authors have declared that no competing interests exist.

times higher than the published GBD 2017 weights for leprosy and the grade 1 disability weight is nearly twenty times higher.

## Conclusions/significance

The global burden of leprosy is grossly underestimated. Revision of the current disability weights and inclusion of disability caused in individuals with grade 0 leprosy disability will contribute towards a more precise estimation of the global burden of leprosy.

## Author summary

Leprosy is a chronic, disabling disease that causes various kinds of disability in the affected person. This includes physical impairment, activity limitation, and participation restriction. However, the published global burden of disease estimates for leprosy is considered to be a gross under-estimation. Disability weights form an integral component in the calculation of the burden estimates. But the methodology for calculation of the weights focuses only on physical impairment and lacks the perspective of the patient. In this study, we systematically reviewed the literature and performed an individual patient data meta-analysis for revising the disability weights for leprosy using domain scores from health-related quality of life instruments. The domains of these instruments cover all aspects of disability from a patient's perspective. We found that the revised weights were considerably higher than the current weights, and were more reflective of the actual disability caused by leprosy. We also found that for individuals without any severe disability due to leprosy (grade 0), they still experience comparable suffering. Revision of the current disability weights and inclusion of the disability caused in grade 0 individuals will contribute towards better estimation of the global burden of leprosy.

## Introduction

Leprosy is a chronic bacterial infection caused by *Mycobacterium leprae*, which may lead to physical disability, stigma, and discrimination. Every year, around 200,000 new leprosy cases are detected worldwide and 12,800 persons are diagnosed with advanced disfigurement (referred to as leprosy grade 2 disability), which is irreversible in nature [1]. Leprosy is often misunderstood to be eliminated or carry no significant disease burden because most cases have limited physical impairment. In reality, the burden of leprosy is beyond physical impairment and encompasses the mental and social wellbeing of affected persons [2, 3]. The World Health Organization (WHO) defines 'disability' as an umbrella term that covers physical impairment, activity limitation, and participation restriction [4]. In leprosy, an alternative definition of disability has traditionally been used; grades 0, 1, and 2 disability [5].

Disease burden can be expressed by Disability Adjusted Life Years (DALYs), a measure that was introduced in 1990 and gained significance with the Global Burden of Disease(GBD) 1996 study [6]. The aim was to holistically quantify disease burden by accounting for morbidity and mortality. DALYs are commonly used to compare the disease burden and set priorities for health policy [7]. The calculation of DALY involves the addition of the Years of Life Lost (YLL) to Years Lived with Disability (YLD) [8]. An important component of the YLD is the pre-calculated disability weight, which has a value ranging from 0 (a state of full health) through 1 (a state equal to death) [9, 10].

It was argued that GBD's quantification of disability weights was based only on the preferences of experts [11, 12]. However, GBD later updated the methodology by including the preferences of general populations in their surveys [8, 9]. It has also been argued that the health state descriptions to calculate disability weights are not accurate for some diseases [11], including leprosy. The descriptions do not always capture all the aspects of disability, such as activity limitation and participation restriction, and sometimes focus only on physical impairment [12]. Therefore the disability weights for leprosy grades 1 and 2 are suspected to be under-estimated for leprosy. Furthermore, no disability weights are assigned in the current GBD study to leprosy patients without any form of physical disability (grade 0), who still may experience poor mental health. The suspected under-estimation of disability weights is flagged by experts [13, 14], but no empirical evidence is available to indicate the magnitude. Improving the health state description will have a direct impact on the global estimation of leprosy burden.

The disability weights can be revised by using Health-Related Quality Of Life (HRQOL) instruments, which cover all three aspects of disability (physical impairment, activity limitation, and participation restriction) equally [15]. HRQOL data can be obtained as (a) Non-Preference based measures, and (b) Preference-based measures [16]. Non-preference based measures describe health states as one or more domains, while preference-based measures assign a 'weight/index value' to the health state from an individual or population, to yield a summary score. For leprosy, the former is the most commonly used method to measure HRQOL. The HRQOL questionnaires can measure the health state from the perspective of the patient and cover all the three aspects of disability.

There is an urgent need to revise the leprosy disability weights for priority setting in health policy, resource allocation, and advocacy. Currently, leprosy is neglected among the Neglected Tropical Diseases (NTD) group and prioritized as the third-least burdened disease [8]. The aim of our study is to compare the current GBD leprosy disability weights with new weights derived using HRQOL measures of leprosy patients. The objectives are: 1) to identify studies that measured HRQOL of leprosy patients; and 2) to assess leprosy disability weights based on the existing HRQOL data. We expect that the revised disability weight will be useful to correct the global leprosy burden. The revised disability weights will also aid to determine the cost-effectiveness of new interventions for treatment in leprosy, which are currently imperative [7].

## Methods

This systematic review and meta-analysis was conducted and reported following the Preferred Reporting Items for Systematic Reviews and Meta-Analyses (PRISMA) statement [17] (S1 Table). First, we searched the International Prospective Register of Systematic Reviews database (PROSPERO) for any similar ongoing or published literature reviews. We then developed a review protocol and registered it with PROSPERO (CRD42019146494).

### Data sources and search strategy

The search strategy (S1 Appendix) was created in collaboration with an information specialist at the Erasmus MC library, with input from the research group. We searched articles on various databases and scrutinized references of the relevant studies. Additionally, we received assistance from the international knowledge center for information resources on leprosy (INFOLEP) [18] in searching for grey literature, such as conference proceedings and inaccessible data. The databases were searched with no restriction on language, year of publication, study design, or setting. We adequately translated non-English publications using online translation tools.

## Study selection

We performed the screening process as a limited dual review. Title and abstracts were first screened by two independent reviewers (SLNC and AT) on a random sample (n = 303) of the study abstracts with clearly defined eligibility criteria (S1 Appendix). An inter-rater agreement of 95% was observed (Kappa = 0.82), which is deemed to be strong [19]. The discrepancies that existed between the reviewers at this stage were resolved through discussion. Based on the discussions at this stage, the screening of the remaining titles and abstracts was adopted by the first reviewer (SLNC). The next stages of screening of the full texts of the selected studies and data-extraction were also carried out by the first reviewer (SLNC). Microsoft Excel spread-sheets and EndNote X9.2 were used for data extraction and management, respectively.

## Data extraction and meta-analysis

Based on the eligibility criteria, we identified studies that used any version of the SF-36 tool, to measure HRQOL in leprosy patients. In September 2019, we sent an invitation to the authors of those studies for joining the Study Consortium. The consortium was set up by including authors who could share the anonymous individual-level data of their study. Primary outcomes of interest were the grade of leprosy disability and eight domain scores of the SF-36 tool.

The Short Form (SF)-36 questionnaire contained 36 items that measured eight health domains: physical functioning, role limitations (physical), bodily pain, general health, vitality, social functioning, role limitations (emotional), and mental health [20]. Aggregated summary scores namely the Physical Component Summary **(PCS)** and the Mental Component Summary (**MCS**), was obtained from the eight domains [21]. Data on age, gender, leprosy type (PB or MB), the grade of disability (Grade 0, 1 or 2), and HRQOL domain scores recorded at baseline for each individual, were used for the analysis. We also confirmed the version of the SF-36 tool used by the authors and checked the scoring methodology for each study.

We imputed any missing data values for each included study separately, using the Multiple Imputation by Chained Reactions (MICE) method. The results of the multiple imputation were investigated using convergence plots (Figs A-D in S2 Appendix). We estimated PCS and MCS scores from the domains of all individual patients, following an oblique factor analysis that allowed for correlation of the underlying factors [22]. We also checked for the sensitivity of the imputation, by re-running the analyses excluding all observations with missing data values.

We transformed the individual SF-36 PCS and MCS scores into equivalent disability weights using a mapping function developed from SF-12 scores (a shortened version of SF-36), by the Institute for Health Metrics and Evaluation (IHME) [23]. The mapping function is based on a loess regression, which returns a disability weight for each composite score (Table B in S2 Appendix). For this, we combined the PCS and MCS scores into a summary score, through simple addition. For a study with multiple imputations, we sorted the estimated disability weights by disability grades for each imputed dataset. Then we pooled the sorted disability weights from each imputation following Rubin's rules [24]. For a study with no imputation, we estimated the mean disability weights, which were also sorted by disability grades.

For meta-analyses, we applied a two-stage multivariate random-effects model to estimate overall mean weights for disability grades 0, 1, and 2 from all consortium studies. Furthermore, we compared the difference between grades 0, 1, and 2 overall disability weights. The inter-study heterogeneity was presented using the $I^2$ statistic. To visualize the results of the meta-analyses we generated forest plots for each grade of disability using a univariate random-effects model. The difference in the estimated weights using the two models were also compared.

In the end, we compared our results to the current disability weights for leprosy, published in the 2017 GBD study. We also used the multivariate meta-analysis approach to estimate overall mean scores for each of the eight SF-36 domains. The meta-analyzed domain scores were sorted by the grade of disability to test for differences in the overall mean domain scores between the grades. We performed all statistical analyses using R version 3.5.3.

## Results

### Literature search

Fig 1 presents the results through the PRISMA flowchart. The literature search yielded 2195 articles. The screening of titles and abstracts resulted in 265 studies that appeared to meet the selection criteria. Following the screening of full text of the included articles, a total of 13 studies were included as they recorded and reported HRQOL scores from the SF-36 tool from people affected by leprosy. One other study reporting SF-36 scores was included from supplementary sources and snowballing procedures while searching for studies published until 2019.

Table 1 presents the study characteristics of the fourteen studies that reported the use of the SF-36 tool. Of these studies, eight were included in further analyses because of the availability of individual-level data (n = 667 patients). In all the studies the proportion of male participants was higher than females. Not all studies reported the type of leprosy of the participants (n = 143) and a few had missing disability grades. No issues were identified on checking the methodology of scoring of SF-36 domains in the included studies. The number of missing data points, if present, were also summarized (S2 Appendix).

### Meta-analysis

Data from eight studies were included in the final meta-analysis. One of the publications [27] included two pilot trials, which are represented separately in the analysis. Fig 2 provides an overview of the overall disability weights for the Grade 2, Grade 1, and Grade 0 disability from the univariate meta-analysis model, for ease of presentation.

Table 2 shows that the differences in the estimated disability weights between the univariate and multivariate meta-analysis models are negligible. But we considered the results of the Multivariate model for reporting in this manuscript because the multivariate model can handle more variability in the analysis than the univariate model. The disability weight for grade 2 was 0.26 (95%CI: 0.18–0.34). For grade 1 disability the estimated weight was 0.19 (95%CI: 0.13–0.26) and grade 0 estimated weight was 0.13 (95%CI: 0.06–0.19). A sensitivity analysis of the imputation showed near similar results (Tables C-D in S2 Appendix).

A test for differences in the estimated disability weights between the three grades of leprosy disability revealed that the weights for grade 1 (0.19) and grade 2 (0.26) were statistically significantly different from grade 0 (0.12)(Table A in S3 Appendix). However, the grade 1 weight was not statistically significantly different from the grade 2 weight.

Table 3 shows the comparison of the estimated disability weights from our multivariate meta-analysis to current 2017 GBD estimates for leprosy [8]. Table 4 shows the current GBD health state description and proposed health state descriptions for all the three grades of leprosy disability, which have been rephrased based on our meta-analyzed HRQOL results.

Fig 3 shows a spider plot of the meta-analyzed domain scores, for each of the disability grades from the multi-variate random-effects model (Table B in S3 Appendix). The Mental Health (MH) mean domain scores were similar in all the three disability grades. While the grade 2 mental health score was not statistically significantly different from grade 0 scores, only marginal statistical significance was observed between the mean mental health scores of

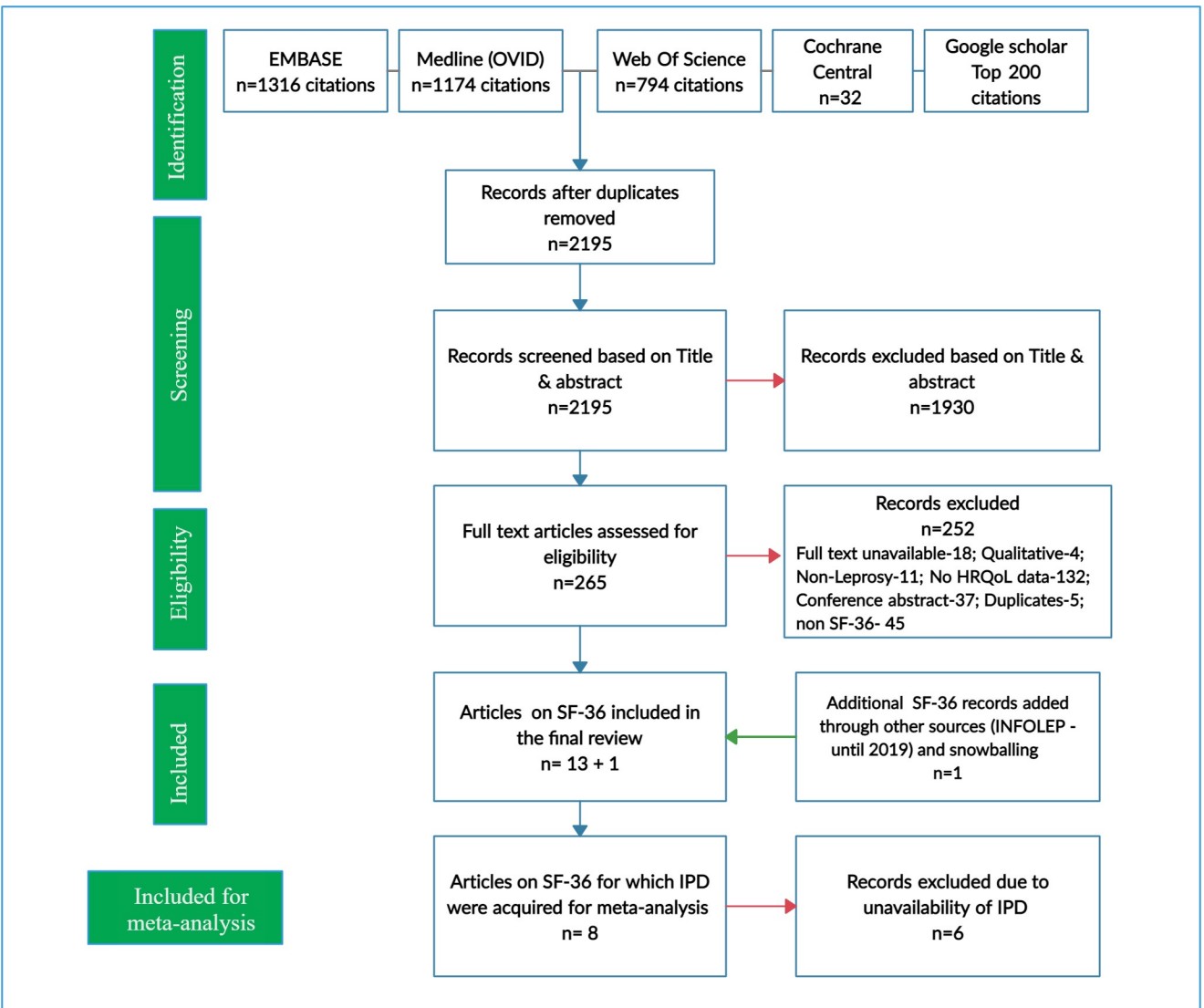

**Fig 1. PRISMA Flowchart.** n- Number of citations, QOL- Quality of Life, IPD- Individual participant data.

grade 0 and grade 1 (Table C in S3 Appendix). For grades 1 and 2, the General Health (GH) and Role limitations due to physical deformity (RP) were the least scored domains with mean scores of 43.70 [95%CI:35–52] and 37.74 [95%CI: 29–47], respectively. The Bodily pain (BP) domain was the second-least scored domain in both grade 1–46.5 [95%CI: 35–58] and grade 2–42.18 [95%CI:31–53] disabilities. However, for grade 0 disability the General Health (GH)-52.28 [95%CI: 44–61] and Bodily Pain (BP) = 54.4 [95%CI: 42–67] were the two least scoring domains.

## Discussion

Our study derived new estimates of leprosy disability weights based on SF-36 HRQOL scores. The overall estimated disability weight for grade 2 disability was 0.26 (95%CI: 0.18–0.34). For grade 1 disability the estimated weight was 0.19 (95%CI: 0.13–0.26) and for grade 0 disability it was 0.13 (95%CI: 0.06–0.19). The revised disability weight for grade 2 leprosy disability is four

**Table 1. Study Characteristic.**

| First Author, Year | HRQL instrument | Country | N | % of Females (number) | Mean Age (Range) | Type of Leprosy |
|---|---|---|---|---|---|---|
| *Demir, et al., 2014 [25] | SF-36 V1 | Turkey | 35 | 31 (11) | 75 (41–96) | PB- 0; MB-35 |
| *Watanabe, H., 2013 [26] | SF-36 V2 | Vietnam | 143 | 40 (57) | 61 (12–89) | Not recorded |
| *#Lambert, et al., 2016 [27] | SF-36 V1 | Ethiopia | 13<br>20 | 23 (3)<br>20 (4) | 30 (18–58)<br>29 (20–45) | PB-0; MB-33 |
| *Lambert, et al., 2016 [28] | SF-36 V1 | Ethiopia | 73 | 21 (15) | 34 (18–60) | PB-0; MB-73 |
| *do Prado, et al., 2011 [29] | SF-36 V1 | Brazil | 97 | 32 (31) | 51 (20–89) | PB-13; MB-84 |
| *Guimenes, et al., 2019 [30] | SF-36 V1 | Brazil | 104 | 44 (46) | 46 (21–80) | PB-20; MB-84 |
| *Bowers, et al., 2017 [31] | RAND-36 | Bangladesh | 75 | 20 (15) | 36 (15–62) | PB-3; MB-72 |
| *Lustosa, et al., 2011 [32] | SF-36 V1 | Brazil | 107 | 37 (40) | 45 (15–86) | PB-50, MB-57 |
| Diaz, et al., 2008 [33] | SF-36 V1 | Brazil | 12 | 58 (7) | 50 | Unavailable |
| Sales, et al., 2017 [34] | SF-36 V1 | Brazil | 59 | 22 (13) | 46 (19–83) | BB-2; BL-16; LL-41 |
| Wan, et al..2017 [35] | RAND-36 | Ecuador | 19 | 37 (7) | 56 | Unavailable |
| Araujo, et al., 2016 [36] | SF-36 V1 | Brazil | 59 | 46 (27) | 46 | PB-11; MB-46 |
| Borges-De-Oliveira, et al., 2015 [37] | SF-36 V1 | Brazil | 126 | 44 (55) | 42 (18–79) | PB-42; MB-84 |
| Bottene, et al., 2012 [38] | SF-36 V1 | Brazil | 49 | 67 (33) | 56 | PB-49; MB-0 |

Study population: N- Sample size, F- Females, PB- Paucibacillary, MB- Multibacillary, RAND- Research and Development, SF- Short Form. The RAND-36 is a publicly available version of the SF-36 that uses a different scoring method.

* Included for meta-analysis.

# Contains two pilot trials that are analyzed as separate datasets.

times higher than the published GBD 2017 weights for leprosy and the grade 1 the disability weight is nearly twenty times higher.

In our study, we determine the under-estimation of the leprosy burden based on empirical evidence. Our meta-analysis is based on data from different time points and geographical settings, hence the external validity is high. Furthermore, the use of individual-level patient data increased the reliability of the estimates. The mapping function used for transforming HRQOL data into disability weights is in line with IHME proposed methods, therefore the acceptance of results can be expected to be high. Finally, the proposed lay descriptions for the disability grades of leprosy were rephrased based on the findings of a reliable and multidimensional patient-reported outcome measure.

As a limitation, we could not include individual data from six of the fourteen identified studies, due to the non-availability of individual data. However, we tried to utilize the HRQOL summary estimates from fourteen studies to understand the overall influence of HRQOL on aspects of disability, i.e., physical impairment, participation restriction, and activity limitation. The derived weights were not corrected for comorbidity, because no data was recorded for comorbidity in most studies. Though, the SF-36 has been designed to provide only a general measure of mental health for most conditions, the use of more sensitive instruments for measuring patient experiences like anxiety and depression [39] would have contributed to more accurate estimates of disability burden in leprosy. Quality appraisal of the included studies was not conducted, because the main focus of the research question was on estimating new weights from the individual participant data that were observed at baseline. Although the studies were from different geographical settings, no study was identified and included from India, which yields the highest number of new cases, globally.

The differences in our disability weights compared to GBD 2017 weights are due to the inclusion of mental health status, activity limitation, and pain, among other effects of leprosy as perceived by the patients. Mental health is one of the lowest scoring domains across all three

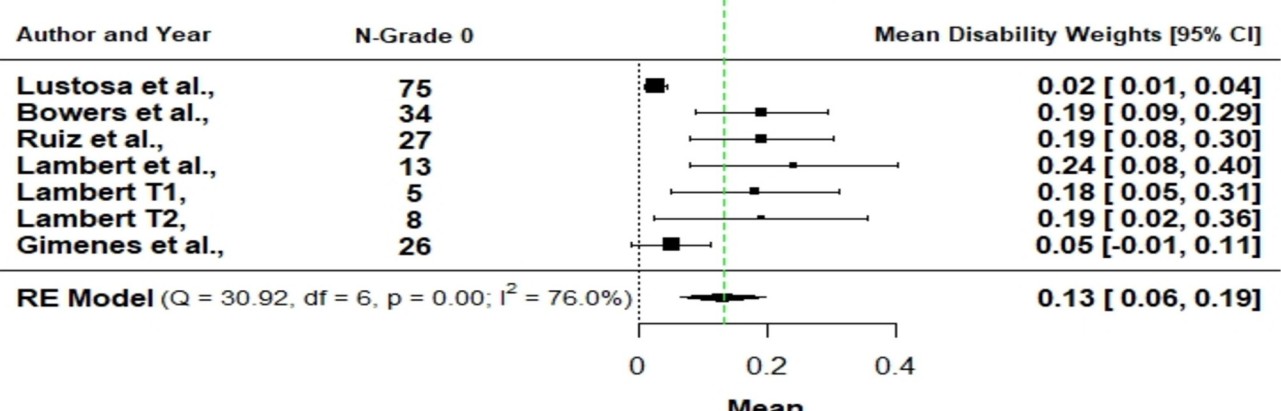

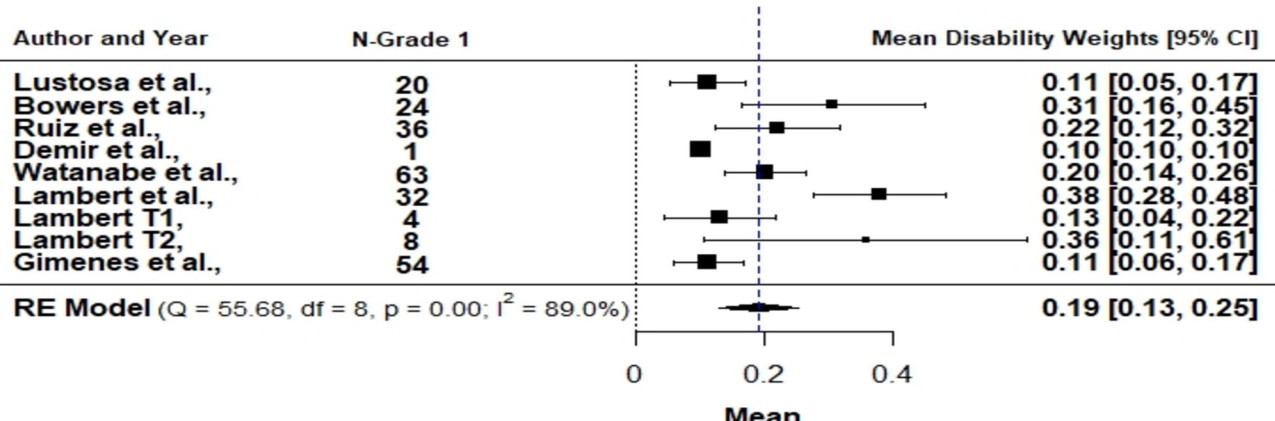

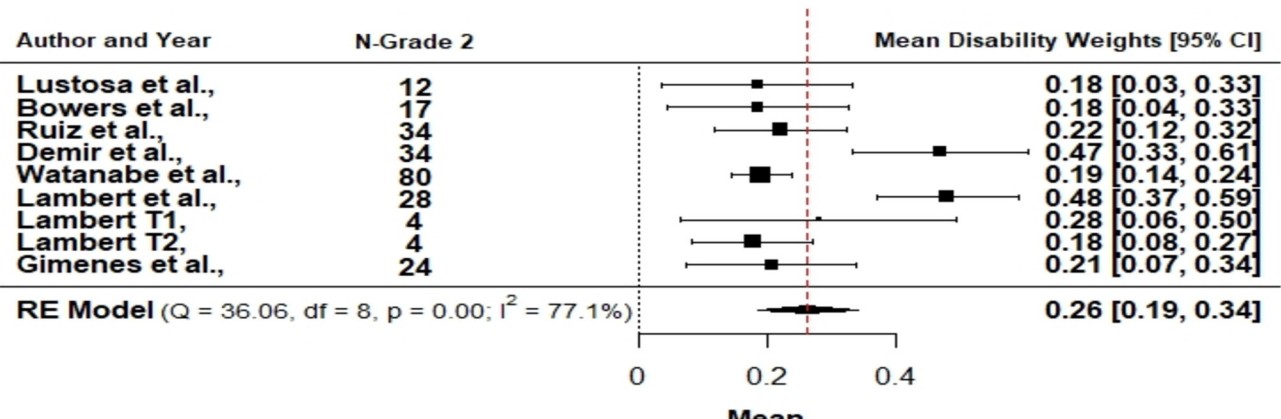

**Fig 2. Forest plots for the three grades of leprosy disability.** N- Number of participants; Plots obtained with the univariate random-effects model.

**Table 2. Meta-Analyses Model Results.**

| Grade | Disability Weight Estimate | | Standard Error | | 95% Confidence Interval | | | |
|---|---|---|---|---|---|---|---|---|
| | Uni | Multi | Uni | Multi | Uni | | Multi | |
| | | | | | lower | upper | lower | upper |
| 0 | **0.13** | **0.12** | 0.03 | 0.03 | 0.06 | 0.20 | 0.06 | 0.19 |
| 1 | **0.19** | **0.19** | 0.03 | 0.03 | 0.13 | 0.25 | 0.13 | 0.26 |
| 2 | **0.26** | **0.26** | 0.04 | 0.04 | 0.19 | 0.34 | 0.18 | 0.34 |

Uni- Two-stage univariate random effects model, Multi- Two-stage random effects multivariate model

grades of leprosy disability. This domain captures how the person feels, including their degree of nervousness and anxiety. Our results are consistent with studies that showed poor mental health in leprosy irrespective of disability. Poor mental health is associated with the high stigma and discrimination experienced by people affected by leprosy for a long time even after the successful completion of antimicrobial multidrug treatment [32, 40, 41]. The role limitations-physical was the least scoring domain among leprosy patients with grade 2 disability, indicating that they face problems in performing regular daily activities as a result of their physical health. A cross-sectional survey by Van Brakel et al. (2012) also described a high proportion of persons with leprosy reporting activity limitations [2]. The same study also showed that limitation in activities increased the risk of social participation restriction and societal discrimination of persons with leprosy. Our results also showed that patients across all three grades of leprosy disability (including grade 0) reported that pain interfered considerably with their normal work. A review by Thakur et al. (2015) also showed that both acute pain associated with leprosy reactions and occurring as intermittent episodes and chronic (neuropathic) pain were common presenting symptoms among leprosy patients [42]. The physical role limitations and bodily pain may have largely contributed to the low scores of the general health domain, for all the grades of leprosy disability.

If the mental health, activity limitation, and bodily pain will be captured sensitively through an appropriate health state description as proposed in our study, then the weights and global burden of leprosy will increase multifold [43]. Modification of disability weights for other NTDs, through the incorporation of the psychological impact of disease has been made in previous GBD studies [44]. Overall, the patient perspective contributed most to the difference in the disability weights. Our revised grade 2 weights are comparable with the GBD 2017 weights of idiopathic-less severe epilepsy (S2 Table). In addition to a general modelling strategy, the GBD study uses different methods for each disease to estimate its global burden. Currently, the leprosy burden is ranked third-last among thirteen NTDs [8], but after considering the revised weights, leprosy is expected to be ranked in the middle order of NTDs.

The early versions of the GBD studies assigned an average disability weight of 0.152 to grades 1 and 2 [45]. The GBD revised its leprosy weights in 2013 [9] and used the same weights in the 2017 burden estimation [8]. The applied methodology was paired comparison of data

**Table 3. Comparison of the revised and GBD 2017 leprosy disability weights.**

| Disability Grade due to Leprosy | Revised Disability Weights | GBD 2017 Disability Weights for leprosy |
|---|---|---|
| Grade 0 –not reported in GBD studies | **0.12** [0.06–0.19] | NR |
| Grade 1 | **0.19** [0.13–0.26] | 0.01 [0.01–0.02] |
| Grade 2 | **0.26** [0.18–0.34] | 0.07 [0.04–0.10] |

NR- Not reported, GBD- Global burden of Disease

**Table 4. GBD 2017 health state descriptions for leprosy.**

| Sequela | GBD health state description for leprosy | Newly proposed health state description for leprosy |
|---|---|---|
| Disfigurement Level 1 due to leprosy (WHO leprosy grade 0) | NR | Has no visible deformity, but skin lesions and mild pain, which causes some mental distress, but no difficulty with daily activities |
| Disfigurement Level 2 due to leprosy (WHO leprosy grade 1) | Has a slight visible deformity that others notice, which causes some worry and discomfort | Has a slight visible deformity, often with moderate pain, which often causes some mental distress, and some difficulty with social and daily activities |
| Disfigurement Level 3 due to leprosy (WHO leprosy grade 2) | Has visible deformity that causes others to stare and comment. As a result, the person is worried and has trouble sleeping and concentrating | Has visible deformity, often with severe pain, which often causes a lot of mental distress, and great difficulty with social and daily activities |

NR- Not Reported, WHO- World Health Organization

collected from the general population, whereas we analyzed SF-36 HRQOL data of individual patients. The difference in methodology is one of the reasons for a drastic difference in the weights. The advantage of HRQOL based weights is that they can sufficiently capture disease burden by covering the patient's perspective [15]. In several instances, GBD considered HRQOL methodology to revise the burden estimates of diseases such as Chronic Obstructive Pulmonary Disease (COPD)and Major Depressive Disorder (MDD) [8].

The revised weights for leprosy disability grade 0 are near similar to grade 1 weights, which implies that the leprosy patients without any visible disability also experience comparable suffering. The proximity in the weights- grade 0 and 1 is largely due to the similarity in the scores of the mental components of the SF-36 tool.

We opted to calculate the PCS and MCS composite scores using oblique factor coefficients [22] over orthogonal factor coefficients [21] because the latter was criticized [46] for its assumption that the two components were independent (uncorrelated). A mapping function was essential to calculate the disability weights from PCS and MCS scores. Therefore, we selected a mapping function developed from SF-12 composite scores proposed by IHME. The SF-12 tool [47] is a shortened version of the more reliable SF-36. The use of the SF-36 composite scores as a proxy to the SF-12 scores is justified because both scores are highly correlated [20, 47].

## Recommendations

We recommend revising the health state descriptions for the two leprosy disability grades because the current description excludes activity limitation and participation restriction aspects that are experienced by the patients. The sequelae base of leprosy should be broadened by including leprosy grade 0 disability patients because its revised weight is close to grade 1. The GBD used a general health state description to calculate the disability weights, which are sensitive to this description. We recommend revising the health state description for leprosy because the current description excludes activity limitation and participation restriction aspects. Finally, there is a need for SF-36 HRQOL studies from India to be included in the future revision of the leprosy disability weights.

## Conclusion

We conclude that the global burden of leprosy is grossly underestimated. Comparable to the patients of grade 2 disability, grade 1 and grade 0 patients also experience a similar level of

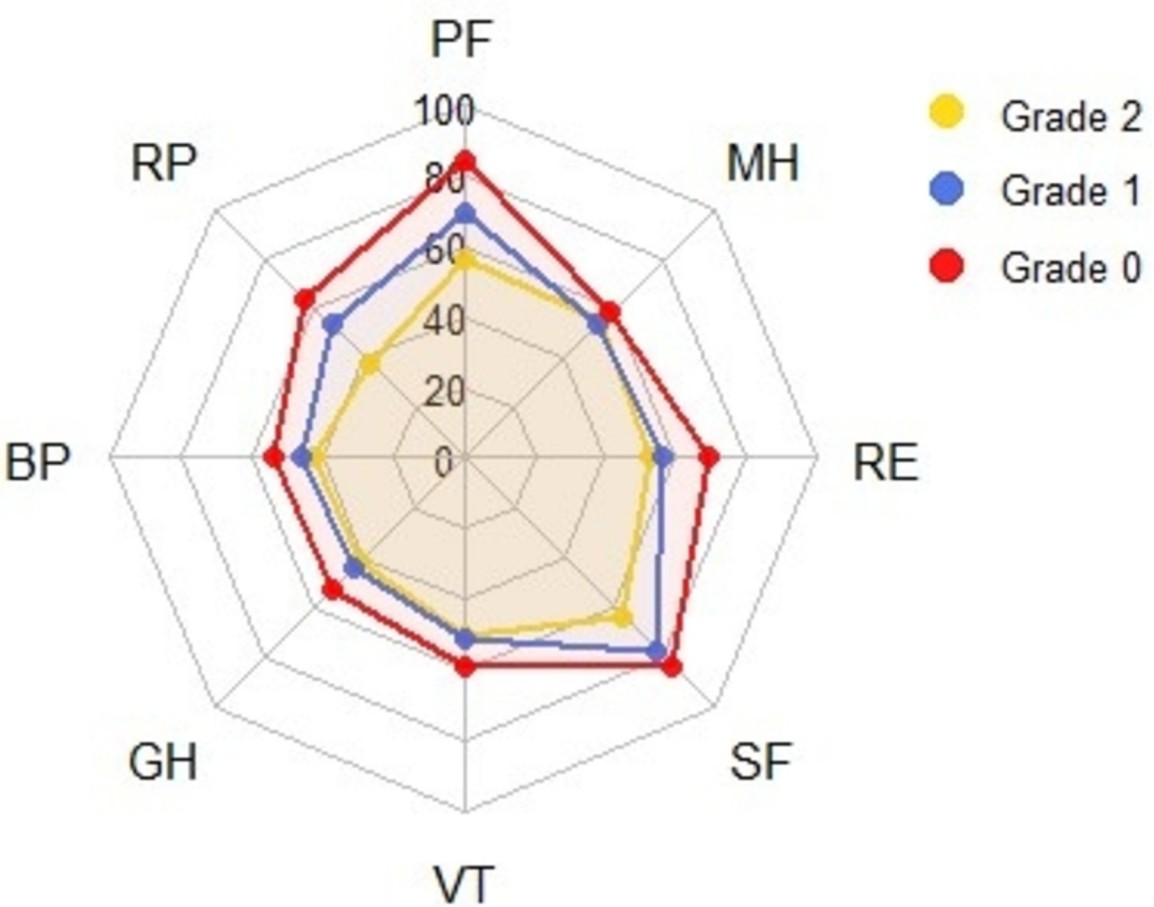

**Fig 3. Spyder plot of reported SF-36 domain scores by the grade of disability.** PF- Physical Function, RP- Role limitations (physical), BP-Bodily Pain, GH- General Health, VT- Vitality, SF- Social Functioning, RE- Role limitations (emotional), MH- Mental Health.

poor mental health and quality of life. Revision of the current disability weights and inclusion of disability caused in individuals with grade 0 leprosy disability will contribute towards a more precise estimation of the global burden of leprosy.

## Supporting information

**S1 Table. PRISMA-IPD checklist.**
(DOCX)

**S2 Table. List of similarly weighted disease sequelae.**
(DOCX)

**S1 Appendix. Search strategy and selection criteria.**
(DOCX)

**S2 Appendix. Summary of missing data, multiple imputation and disability weight transformation.**
(DOCX)

**S3 Appendix. Summary of meta-analyzed domain scores.**
(DOCX)

## Acknowledgments

We thank Sabrina Gunput, biomedical information specialist at the Erasmus MC Library, for her technical assistance in the development of the search strategy for the review and Daan Nieboer, Department of Public Health, Erasmus MC for his statistical advice. We also thank Jiske Erlings at Infolep-Amsterdam, for her assistance with inaccessible studies during the review phase, and Eliane Ignotti for her assistance in contacting the authors.

## Author Contributions

**Conceptualization:** Shri Lak Nanjan Chandran, Anuj Tiwari, Juanita Haagsma, Jan Hendrik Richardus.

**Data curation:** Shri Lak Nanjan Chandran, Anuj Tiwari.

**Formal analysis:** Shri Lak Nanjan Chandran, Anuj Tiwari.

**Investigation:** Shri Lak Nanjan Chandran, Anuj Tiwari.

**Methodology:** Shri Lak Nanjan Chandran, Anuj Tiwari, Juanita Haagsma, Jan Hendrik Richardus.

**Project administration:** Shri Lak Nanjan Chandran, Anuj Tiwari.

**Resources:** Shri Lak Nanjan Chandran, Anuj Tiwari, Anselmo Alves Lustosa, Betul Demir, Bob Bowers, Rachel Gimenes Rodrigues Albuquerque, Renata Bilion Ruiz Prado, Saba Lambert, Hiroyuki Watanabe, Juanita Haagsma, Jan Hendrik Richardus.

**Software:** Shri Lak Nanjan Chandran, Anuj Tiwari.

**Supervision:** Jan Hendrik Richardus.

**Validation:** Anuj Tiwari, Juanita Haagsma, Jan Hendrik Richardus.

**Visualization:** Shri Lak Nanjan Chandran, Juanita Haagsma, Jan Hendrik Richardus.

**Writing – original draft:** Shri Lak Nanjan Chandran.

**Writing – review & editing:** Anuj Tiwari, Anselmo Alves Lustosa, Betul Demir, Bob Bowers, Rachel Gimenes Rodrigues Albuquerque, Renata Bilion Ruiz Prado, Saba Lambert, Hiroyuki Watanabe, Juanita Haagsma, Jan Hendrik Richardus.

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
