## [Decision Letter · Decision Letter 0]

4 Jan 2021

Dear Dr. Richardus,

Thank you very much for submitting your manuscript "Revised estimates of leprosy disability weights for assessing the global burden of disease: A systematic review and individual patient data meta-analysis" for consideration at PLOS Neglected Tropical Diseases. As with all papers reviewed by the journal, your manuscript was reviewed by members of the editorial board and by several independent reviewers. The reviewers appreciated the attention to an important topic. Based on the reviews, we are likely to accept this manuscript for publication, providing that you modify the manuscript according to the review recommendations. 

Sincerely,

Linda B Adams

Associate Editor

Gerson Penna

Deputy Editor

Reviewer's Responses to Questions

**Key Review Criteria Required for Acceptance?**

**Methods**

-Are the objectives of the study clearly articulated with a clear testable hypothesis stated?

-Is the study design appropriate to address the stated objectives?

-Is the population clearly described and appropriate for the hypothesis being tested?

-Is the sample size sufficient to ensure adequate power to address the hypothesis being tested?

-Were correct statistical analysis used to support conclusions?

-Are there concerns about ethical or regulatory requirements being met?

Reviewer #1: Yes to all questions above

Reviewer #2: This is a well defined and organized study

**Results**

-Does the analysis presented match the analysis plan?

-Are the results clearly and completely presented?

-Are the figures (Tables, Images) of sufficient quality for clarity?

Reviewer #1: Table 1: No explanation is given for the acronym ‘RAND-36’ or its equivalence with the SF-36

Table 5: People with Grade 1 disabilities also face participation restrictions and suffer from depression. Grade 2 disability is a frequent cause of depression and of severe participation restrictions. A recent systematic review found that up to 50% of persons affected by leprosy suffer from depression and/or anxiety. This is insufficiently captured in the proposed health state descriptions.

Reviewer #2: Results are clear and well analyzed

**Conclusions**

-Are the conclusions supported by the data presented?

-Are the limitations of analysis clearly described?

-Do the authors discuss how these data can be helpful to advance our understanding of the topic under study?

-Is public health relevance addressed?

Reviewer #1: Line 358-65: the authors are advised to consider rephrasing the proposed health state descriptions since ‘daily activities’ is often taken to mean activities of daily living, i.e. the activity component of the ICF, rather than the participation component. Furthermore, an explicit reference to poor mental wellbeing (or mental health) is recommended.

Reviewer #2: The conclusions are logical and supported by the analysis

**Editorial and Data Presentation Modifications?**

Reviewer #1: Discussion

I miss a discussion of the impact of the number of people counted in the DALY calculation (based on new cases only?) and the duration of the disability (only counted during MDT?). If these are not included appropriately, this will also significantly affect the resulting estimates.

Line 313-4: This may indicate that the SF-36 is not the most appropriate instrument to quantify the experience of depression and anxiety, which are found to be very common among persons affected by leprosy. It would be appropriate to note this in the light of the findings in the below two studies.

Line 316-8: see also Somar et al (2020). The impact of leprosy on the mental wellbeing of leprosy-affected persons and their family members - a systematic review.

Also Van Dorst et al (2020): ref 3 and 

Bow-Bertrand et al (2019). An exploration into the psychological impact of leprosy in Sirajganj, Bangladesh.

Line 332: It may be worth noting that this has been found also for other NTDs, e.g.

Bailey F, Mondragon-Shem K, Haines LR, Olabi A, Alorfi A, Ruiz-Postigo JA, Alvar J, Hotez P, Adams ER, Vélez ID, Al-Salem W. Cutaneous leishmaniasis and co-morbid major depressive disorder: A systematic review with burden estimates. PLoS neglected tropical diseases. 2019 Feb 25;13(2):e0007092.

Bailey F, Eaton J, Jidda M, van Brakel WH, Addiss DG, Molyneux DH. Neglected tropical diseases and mental health: Progress, partnerships, and integration. Trends in parasitology. 2019 Jan 1;35(1):23-31.

Line 343: I would challenge this statement. HRQOL based weights may do a better job than the previous methods, but the SF-36 may still underestimate the impact, especially on social and work participation and on mental health, since it has not been designed to measure these based on the ICF and DSM frameworks. The very common impact on family members is not taken into account either.

Reviewer #2: Probably has too many tables and seems overly technical

**Summary and General Comments**

Reviewer #1: The DALY weights for leprosy are an important topic since revision has been overdue for a long time!

Reviewer #2: The authors present here a meta-analysis of leprosy disability weighting estimates, with particular emphasis on patient input. They suggest that leprosy is among the more “neglected of the neglected tropical diseases” in that the current antiquated disability estimates erroneously suggest a false low burden of disease in most populations, that can incorrectly skew distribution of health resources or bely the benefits of new intervention programs. This results mainly from a lack of patient input on psychological and social issues and a traditional reliance of physical impairment scores. 

It is mind boggling that a disease so widely known, feared and stigmatized around the world, would not somehow include patient input on mental health and social impacts of their disease. Yet, among the more than 2000 manuscripts reviewed in this analysis only 14 included standardized patient input and among those only 8 also had robust enough data to be re-analyzed in this study. Perhaps not surprisingly, they observe that individuals with grade 0 and 1 disabilities also suffer mental health and social deficiencies and the resulting true burden of those disabilities are 4-20 times greater than previously understood.

This is an important contribution to bring leprosy more into the modern world. Based on solid statistical analysis the authors responsibly offer new disability definitions that can change the paradigm of disability estimation and disease burden in leprosy. The language of the paper is somewhat technical and the total number of tables could easily be halved. 

In revision the authors should consider:

Line 222: “a” few

Lines 346-349: poorly phrased. As written implies the revised weights are not different from before. Should say they 'are similar to each other'

PLOS authors have the option to publish the peer review history of their article (what does this mean?). If published, this will include your full peer review and any attached files.

Reviewer #1: Yes: Wim H van Brakel

Reviewer #2: No
---

## [Editor Report · Decision Letter 1]

5 Feb 2021

Dear Drs. Richardus and Chandran,

We are pleased to inform you that your manuscript 'Revised estimates of leprosy disability weights for assessing the global burden of disease: A systematic review and individual patient data meta-analysis' has been provisionally accepted for publication in PLOS Neglected Tropical Diseases.

Before your manuscript can be formally accepted you may need to complete some formatting changes, which you will receive in a follow up email. A member of our team will be in touch with a set of requests.

Best regards,

Linda B Adams

Associate Editor

Gerson Penna

Deputy Editor

---

## [Editor Report · Acceptance letter]

22 Feb 2021

Dear Prof. Richardus,

We are delighted to inform you that your manuscript, "Revised estimates of leprosy disability weights for assessing the global burden of disease: A systematic review and individual patient data meta-analysis," has been formally accepted for publication in PLOS Neglected Tropical Diseases.

Best regards,

Shaden Kamhawi

co-Editor-in-Chief

Paul Brindley

co-Editor-in-Chief
